# Attitude and beliefs about the social environment associated with chemsex among MSM visiting STI clinics in the Netherlands: An observational study

Ymke J. Evers[ID][1,2☯]*, Jill J. H. Geraets[1☯], Geneviève A. F. S. Van Liere[1,2], Christian J. P. A. Hoebe[1,2], Nicole H. T. M. Dukers-Muijrers[1,2]

1 Department of Sexual Health, Infectious Diseases and Environmental Health, Public Health Service South Limburg, Heerlen, The Netherlands, 2 Department of Social Medicine and Medical Microbiology, Care and Public Health Research Institute (CAPHRI), Maastricht University Medical Center, Maastricht, The Netherlands

☯ These authors contributed equally to this work.
* ymke.evers@ggdzl.nl

## Abstract

### Background

Drug use during sex, 'chemsex', is common among men who have sex with men (MSM) and related to sexual and mental health harms. This study assessed associations between chemsex and a wide range of determinants among MSM visiting STI clinics to increase understanding of characteristics and beliefs of MSM practicing chemsex.

### Methods

In 2018, 785 MSM were recruited at nine Dutch STI clinics; 368 (47%) fully completed the online questionnaire. All participants reported to have had sex in the past six months. Chemsex was defined as using cocaine, crystal meth, designer drugs, GHB/GBL, ketamine, speed or XTC/MDMA during sex in the past six months. Associations between chemsex and psychosocial determinants, socio-demographics, sexual behaviour and using tobacco or alcohol were assessed by multivariable logistic regression analyses.

### Results

Chemsex was reported by 44% of MSM (161/368) and was not associated with socio-demographics. Independent determinants were 'believing that the majority of friends/sex partners use drugs during sex' (descriptive norm) (aOR: 1.95, 95%CI: 1.43–2.65), 'believing that sex is more fun when using drugs' (attitude) (aOR: 2.06, 95%CI: 1.50–2.84), using tobacco (aOR: 2.65, 95%CI: 1.32–5.32), multiple sex partners (aOR: 2.69, 95%CI: 1.21–6.00), group sex (aOR: 4.65, 95%CI: 1.54–14.05) and using online dating platforms (aOR: 2.73, 95%CI: 1.13–6.62).

**Data Availability Statement:** Data in this study contain potentially identifying and sensitive patient information. Due to the General Data Protection

Regulation, it is not allowed to distribute or share any personal data that can be traced back (direct or indirect) to an individual. Moreover, publicly sharing the data would not be in accordance with participant consent for this study. Therefore, interested researchers should contact the Medical Ethical Committee of the University of Maastricht and the head of the data-archiving of the Public Health Service South Limburg (Helen Sijstermans: helen.sijstermans@ggdzl.nl) when they would like to re-use data.

**Funding:** The authors received no specific funding for this work.

**Competing interests:** The authors have declared that no competing interests exist.

## Conclusion

MSM are likely to find themselves in distinct social networks where it is the norm to use drugs when having sex and pleasure is linked to chemsex. Health services should acknowledge the social influence and pleasurable experiences to increase acceptability of strategies aimed at minimizing the possible harms of chemsex.

## Introduction

Drug use during sex, or 'chemsex', is a well-documented behaviour among men who have sex with men (MSM) [1,2]. The main type of drugs described in chemsex are drugs that stimulate sexual desire, such as crystal methamphetamine, gamma-hydroxybutyric acid (GBH)/gamma-butyrolactone (GBL), mephedrone, ecstasy (XTC) and ketamine [3,4]. A systematic review that included studies from Europe, Australia, and the United States of America, demonstrated that the proportion of MSM practicing chemsex ranged from 17% to 27% among MSM attending STI clinics in large cities [1]. A recent study by our research group showed that chemsex was also common practice among MSM living outside large cities in the Netherlands (36%) [5].

The upsides of chemsex that have been described for MSM are various and include the enhancement of sexual desire [6,7]. Chemsex has been associated with greater sexual satisfaction, but lower life satisfaction [8]. Chemsex has been associated with several adverse sexual and mental health outcomes. For example, chemsex has been associated with an increased risk of condomless anal intercourse [9–12], acquisition of sexually transmitted infections (STIs) [10,13,14] and anxiety and depression when the use was considered dependent [15].

Considering that chemsex is common among MSM and associated with health harms, health promotion strategies aimed at minimizing these harms, such as STI control strategies and safer drug use (harm reduction) strategies are important in this group. An understanding of the characteristics and beliefs associated with chemsex is needed to identify MSM practicing chemsex in health care and tailor health promotion strategies to their beliefs. Previous data on associations between chemsex and socio-demographics, i.e. age, ethnicity and educational level, have been inconsistent [1,8,14,16–18]. Concerning the use of other substances, tobacco use has been associated with chemsex, but no association has been found between regular alcohol consumption and chemsex [14,15]. Sexual risk behaviour, i.e. multiple sex partners, casual sex partners, group sex, and fisting, has been associated with chemsex [8,10,13,18–20]. A few studies have assessed associations between chemsex and psychosocial determinants from common behaviour-oriented theories (e.g. Theory of Planned Behaviour, Health Belief Model, Social Norms Theory). Lower perception of risk, or 'risk perception' [21], was associated with drug use and regular drug use in MSM [22]. A more favourable appraisal, or 'attitude' [23], towards drug use among lesbian, gay and bisexual people helped to explain the higher use of drugs compared to heterosexuals [24]. Acceptability of drug use among the social environment, or the 'subjective norm' [25], was positively associated with drug use in MSM [22]. Believing that most people in the social environment practice chemsex, or the 'descriptive norm', was also related to chemsex among MSM in a qualitative study [19]. Adequate ability to refuse drugs and confidence in this ability, or 'refusal skills and self-efficacy' [26], were associated with less drug use among MSM [27]. Finally, a qualitative study found that drug use could be the result of a learned automatic response, or 'habit' [28], to sex for some MSM [29].

Most studies focused on a particular set of determinants. Only one quantitative study [22] assessed a broad framework of the above described psychosocial determinants, socio-demographics and sexual risk behaviour in relation with drug use among MSM. However, this study [22] assessed determinants of drug use in general, and these determinants might differ for chemsex. Therefore, to explore characteristics and beliefs of MSM practicing chemsex, in the present quantitative study, we developed a comprehensive framework including psychosocial determinants, socio-demographics, tobacco and alcohol use, and sexual behaviour possibly associated with chemsex (Fig 1).

## Methods and materials

### Ethics statement

This study was approved by the Medical Ethical Committee of the University of Maastricht (METC 2018–0485). STI client registry data was collected within standard care. During recruitment, STI nurses provided information on the study and participants gave oral informed consent to be approached for the online questionnaire and this was registered in the STI client registry. Participants gave written informed consent at the beginning of the questionnaire (S1 File). Data from questionnaires and linkage to data from medical records were retrieved in a coded de-identified manner.

### Study design

The outpatient Public Health Service STI clinics in the Netherlands offer free and anonymous STI and HIV testing, hepatitis B vaccinations, and sexual health counselling for risk groups, such as MSM. Of all unique clients visiting any STI clinic in the Netherlands in 2018 (111,271), 27% (29,531) were MSM [30]. MSM was defined as a man who reported having sex with a man in the preceding six months. Nurses of nine STI clinics in the Netherlands (list of participating STI clinics in S2 File) were instructed to recruit MSM aged 16 years or older during their STI consultations for participation in an online questionnaire (S3 File) on drug use; it was explicitly instructed to include MSM regardless of their drug use (yes or no) behaviour. Convenience sampling was used for recruitment. The recruitment period was three months in 2018 for all participating STI clinics.

### Procedures

All STI consultations included a standardized nurse-taken medical and sexual history, including socio-demographics and sexual behaviour in the past six months. These data were registered in an electronic patient registry. Up to one week after the STI consultation, the questionnaire was sent by email to all MSM who had agreed to participate in the study. It was made clear that the questionnaire was intended for MSM regardless whether they used drugs or not in the past six months. The questionnaire included questions on drug use during sex, sexual behaviour, alcohol and tobacco use, and psychosocial determinants. Questionnaire data were linked to electronic patient registry data using the consultation code.

### Outcome

The outcome in this study was chemsex, the use of drugs before or during sex, in the past six months. Crystal meth, cocaine, designer drugs (2-CB, 3 MMC, 4-FA, 4-MEC), GHB, GBL, ketamine, MDMA, mephedrone, speed, and XTC were included in this definition. Alcohol and cannabis are generally excluded from the chemsex definition because of their common use in a recreational context. Poppers are used during sex by MSM, but only using poppers during sex is normally not classified as chemsex in previous literature [3,9–11,13].

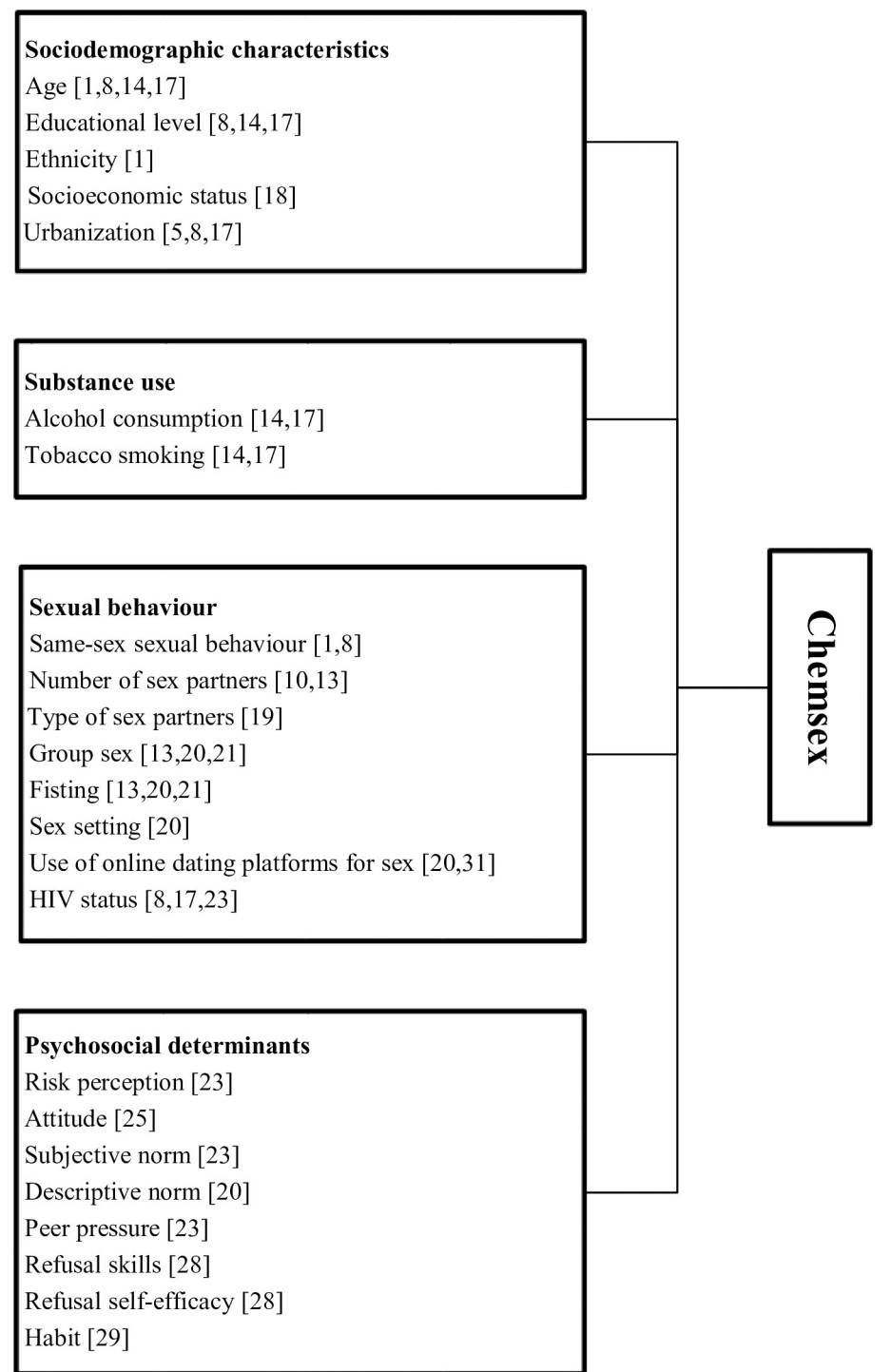

**Fig 1. Conceptual framework to explain the use of drugs during sex (chemsex) in men who have sex with men attending Dutch sexually transmitted infection clinics.**

### Determinants

**Socio-demographics.** Age, four-digit postal codes, educational level, and ethnicity were available from the coded electronic patient registry. Socioeconomic status (SES) scores based

on income, educational level and employment were extracted from the Netherlands Institute for Social Research (http://www.scp.nl) per four-digit postal code area. Postal codes were also used to calculate urbanization (www.//cbs.nl), urbanization was categorized into urban ($\geq$1500 addresses/m$^2$) and non-urban ($<$1500 addresses/ m$^2$). Categories for educational level (practical, theoretical) and ethnicity (western, non-western) were based on the definitions used by the Central Bureau of Statistics (www.//cbs.nl).

**Substance use.**   Frequency of alcohol consumption was coded into at least two times per week and four times per month or less. Tobacco use was measured as current smoking (yes or no).

**Sexual behavior.**   Number of sex partners in the past six months, bisexual behaviour (yes, no), and HIV status were available from the coded electronic patient registry. Type of sex partners was categorized into having sex with a steady partner, if participants reported only having sex with a steady partner, and having sex with casual partner(s), if participants reported having sex with other than steady partners (as well). Sex at the party scene included having reported to have sex at sex parties, darkrooms, or after parties. The use of online dating platforms included having reported to use dating apps or websites to find sex partners.

**Psychosocial determinants.**   The chosen list of items measuring the psychosocial determinants included a range of beliefs about the risks, advantages, social environment, refusal skills and habitual thoughts. The two items for descriptive norm had Cronbach's alpha higher than 0.8 and were combined in a mean score (Table 1).

## Statistical analyses

Participants were defined as all MSM who fully completed the questionnaire. Descriptive analyses were used to calculate the proportion of participants reporting chemsex and proportions

**Table 1.  Measurement of psychosocial determinants.**

| Determinants | Questions | Answering scale | Answering options | Interpretation | Items | Internal consistency |
|---|---|---|---|---|---|---|
| Risk perception | *I think that I am more at risk of acquiring an STI when I use drugs during sex* | Five-point Likert Scale | Totally disagree–totally agree | A higher score indicates having a higher risk perception of the consequences of chemsex | 1 | NA |
| Attitude | *I think that I can enjoy sex more when I use drugs* | Five-point Likert Scale | Totally disagree–totally agree | A higher score indicates a more positive attitude towards chemsex | 2 | 0.38 |
| | *I think that my fear of acquiring an STI decreases when I use drugs during sex* | | | | | |
| Subjective norm | *My friends consider using drugs during sex to be a fun thing to do* | Five-point Likert Scale | Totally disagree–totally agree | A higher score indicates that chemsex is more considered to be a fun thing to do | 1 | NA |
| Descriptive norm | *How many of your friends use drugs during sex?* | Five-point Likert Scale | No one–everyone | A higher score indicates that more friends use drugs during sex | 2 | 0.83 |
| | *How many of your sex partners use drugs during sex?* | | | | | |
| Peer pressure | *I sometimes experience pressure from my friends or sex partners to use drugs during sex* | Five-point Likert Scale | Totally disagree–totally agree | A higher score indicates more peer pressure | 1 | NA |
| Refusal skills & self-efficacy | *I know how to refuse drugs during sex if I do not want to use them* | Five-point Likert Scale | Totally disagree–totally agree | A higher score indicates more refusal skills & self-efficacy | 2 | 0.61 |
| | *I manage to say no to friends or sex partners who offer me drugs during sex* | | | | | |
| Habit | *I automatically think about drugs when having sex* | Five-point Likert Scale | Totally disagree–totally agree | A higher score indicates a more habitual thought | 1 | NA |

NA: Not applicable.

of participants reporting specific psychosocial beliefs about chemsex. We then assessed determinants for chemsex. We started with comparing socio-demographics, alcohol/tobacco use, sexual behaviour, and psychosocial determinants between MSM practicing chemsex and MSM not practicing chemsex using chi-square tests and t-tests (for continuous variables). Associations between chemsex and all variables were then assessed using univariable logistic regression analyses. Then, we built a multivariable four-step hierarchical regression model. The four blocks consisted of variables that were significantly (p<0.05) associated with chemsex in univariable analyses (as shown in Table 2). In the first step, sociodemographics were entered (block 1). In the second step, tobacco/alcohol use variables were entered (block 2). In the third step, sexual behaviour variables were entered (block 3). Finally, psychosocial determinants were entered in the regression model (block 4). No variables were removed during the steps. For each block, the explained variance was described. The order of block-entry was based on previous literature (as described in Fig 1), known categories of variables associated with chemsex were entered first and the new category of variables, i.e. psychosocial determinants, were entered in the final step. As this study concerns an exploratory study, we did not adjust for multiple comparisons. All analyses were performed using IBM SPSS Statistics Version 24.

## Results

### Study population

A total of 785 MSM were recruited to participate during the study period and 368 (47%) fully completed the questionnaire. Participants had a median age of 40 years (IQR 32–47), 65% was higher educated and 93% had a western nationality. The median number of sex partners in the preceding six months was 6 (IQR: 4–10). Chemsex in the past six months was reported by 44% (161/368) of participants, of whom 47% had chemsex in the last week. Among participants practicing chemsex, the most common drugs used were XTC/MDMA (85%), GHB/GBL (78%), and ketamine (43%). Crystal meth was reported by 9% and mephedrone by 4%. The use of four or more different drugs in the past six months was reported by 42% of MSM practicing chemsex. Combining different drugs during one chemsex session was reported by 81%; XTC/MDMA and GHB/GBL was the most reported combination (75%). Intravenous injection of drugs was reported by 4%. Among all participants, tobacco use was reported by 38% and regular alcohol consumption by 47%. The majority reported to have sex with casual partners (79%) and used online dating platforms for sex (78%). In general, participants reported to be aware of the STI risks of chemsex (median risk perception: 3.5), managed to refuse drugs during sex, (median refusal self-efficacy: 4.4), and did not experience peer pressure to use drugs during sex (median peer pressure: 1.6). Fig 2 describes the proportions for the specific categories of psychosocial items.

### Determinants associated with chemsex: Univariable analyses

MSM who reported chemsex more often believed that they could enjoy sex more when using drugs (mean attitude 3.9 vs. 2.2, p<0.001), felt less skilled to refuse drugs (mean refusal skills: 4.5 vs. 4.6, p = 0.02), perceived greater acceptance (mean subjective norm: 3.7 vs. 2.8, p<0.001) and performance (mean descriptive norm: 3.7 vs. 2.8, p<0.001) of chemsex among their social environment, and scored higher on automatically thinking about drugs when having sex (mean habitual thought: 2.2 vs. 1.3, p<0.001) compared to MSM who did not report chemsex. Table 2 shows all other variables that were significantly associated with chemsex in univariable analyses.

**Table 2. Characteristics of study population and univariable analyses of associations between chemsex and sociodemographics, alcohol/tobacco use, sexual behaviour and psychosocial determinants.**

| | All participants (N = 368) | Chemsex MSM (N = 161) | No chemsex MSM (N = 207) | Outcome: chemsex |
|---|---|---|---|---|
| | % of total (N) or mean ± SD | % within groups (N) or mean ± SD | % within groups (N) or mean ± SD | OR (95% CI) |
| **Sociodemographics** | | | | |
| Age | 40.6 ± 13.6 | 43 ± 12 | 38 ± 14 | **1.03 (1.01–1.04)**\*\* |
| Ethnicity | | | | |
| Western | 92.9 (342) | 43.6 (149) | 56.4 (193) | 1 |
| Non-western | 7.1 (26) | 46.2 (12) | 53.8 (14) | 1.11 (0.50–2.47) |
| SES† | | | | |
| Low | 33.4 (123) | 48.8 (60) | 51.2 (63) | 1 |
| Middle | 32.6 (120) | 35.0 (42) | 65.0 (78) | **0.57 (0.34–0.95)**\* |
| High | 34.0 (125) | 47.2 (59) | 52.8 (66) | 0.94 (0.57–1.55) |
| Educational level | | | | |
| Practical | 35.1 (129) | 47.3 (61) | 52.7 (68) | 1 |
| Theoretical | 64.9 (239) | 41.8 (100) | 58.2 (139) | 0.80 (0.52–1.23) |
| Urbanization | | | | |
| Non-urban | 44.8 (165) | 40.6 (67) | 59.4 (98) | 1 |
| Urban | 55.2 (203) | 46.3 (94) | 53.7 (109) | 1.26 (0.83–1.91) |
| **Alcohol/tobacco use** | | | | |
| Tobacco smoking | | | | |
| No | 62.2 (229) | 56.8 (79) | 43.2 (60) | 1 |
| Yes | 37.8 (139) | 35.8 (82) | 64.2 (147) | **2.36 (1.53–3.63)**\*\*\* |
| Regular alcohol consumption | | | | |
| No | 52.7 (194) | 43.7 (76) | 56.3 (98) | 1 |
| Yes | 47.3 (174)_ | 43.8 (85) | 56.2 (109) | 0.99 (0.66–1.50) |
| **Sexual behaviour** | | | | |
| Same-sex sexual behaviour | | | | |
| Only men | 89.9 (331) | 41.7 (138) | 58.3 (193) | 1 |
| Men and women | 10.1 (37) | 62.2 (23) | 37.8 (14) | **2.30 (1.14–4.62)**\* |
| Type of sex partner(s) | | | | |
| Steady | 20.7 (76) | 34.2 (26) | 65.8 (50) | 1 |
| Casual | 79.3 (292) | 46.2 (135) | 53.8 (157) | 1.65 (0.98–2.80) |
| Number of sex partners past six months | | | | |
| <5 | 38.3 (141) | 26.2 (37) | 73.8 (104) | 1 |
| 5–10 | 39.4 (145) | 52.4 (76) | 47.6 (69) | **3.10 (1.88–5.09)**\*\*\* |
| >10 | 22.3 (82) | 58.5 (48) | 41.5 (34) | **3.97 (2.23–7.07)** \*\*\* |
| Number of sex partners during sex | | | | |
| 1 | 67.1 (247) | 30.0 (74) | 70.0 (173) | 1 |
| 2–3 | 22.3 (82) | 73.2 (60) | 26.8 (22) | **6.38 (3.65–11.15)**\*\*\* |
| >3 | 10.6 (39) | 69.2 (27) | 30.8 (12) | **5.26 (2.53–10.94)**\*\*\* |
| Group sex | | | | |
| No | 81.8 (301) | 34.9 (105) | 65.1 (196) | 1 |
| Yes | 18.2 (67) | 83.6 (56)) | 16.4 (11) | **9.50 (4.77–18.92)**\*\*\* |
| Fisting | | | | |
| No | 91.0 (335) | 39.7 (133) | 60.3 (202) | **1** |

*(Continued)*

**Table 2.** (Continued)

| | All participants (N = 368) | Chemsex MSM (N = 161) | No chemsex MSM (N = 207) | Outcome: chemsex |
|---|---|---|---|---|
| | % of total (N) or mean ± SD | % within groups (N) or mean ± SD | % within groups (N) or mean ± SD | OR (95% CI) |
| Yes | 9.0 (33) | 84.8 (28) | 15.2 (2) | **8.51 (3.20–22.58)***** |
| Sex at the party scene | | | | |
| No | 74.5 (274) | 35.4 (97) | 64.5 (177) | 1 |
| Yes | 25.5 (94) | 68.1 (64) | 31.9 (30) | **3.89 (2.36–6.41)***** |
| Use of online dating platforms | | | | |
| No | 22.0 (81) | 28.4 (23) | 71.6 (58) | 1 |
| Yes | 78.0 (287) | 48.1 (138) | 51.9 (149) | **2.34 (1.37–3.99)**** |
| Known positive HIV status | | | | |
| No | 89.7 (330) | 39.7 (131) | 60.3 (199) | 1 |
| Yes | 10.3 (38) | 78.9 (30) | 21.1 (8) | **5.70 (2.53–12.81)***** |
| **Psychosocial determinants‡** | | | | |
| Risk perception | 3.5 ± 1.2 | 3.5 ± 1.2 | 3.5 ± 1.2 | 0.97 (0.81–1.15) |
| Attitude (enjoyment) § | 2.9 ± 1.2 | 3.9 ± 1.1 | 2.2 ± 1.2 | **3.45 (2.68–4.44)***** |
| Attitude (decreased STI anxiety) ¶ | 2.3 ± 1.1 | 2.4 ± 1.2 | 2.2 ± 1.2 | 1.19 (1.00–1.42) |
| Refusal skills | 4.5 ± 0.9 | 4.5 ± 1.0 | 4.6 ± 0.8 | **0.74 (0.68–0.95)*** |
| Refusal self-efficacy | 4.4 ± 1.0 | 4.3 ± 1.0 | 4.5 ± 1.0 | 0.86 (0.70–1.05) |
| Subjective norm | 3.2 ± 1.1 | 3.7 ± 0.9 | 2.8 ± 1.1 | **2.66 (2.05–3.44)***** |
| Descriptive norm | 2.6 ± 1.5 | 3.7 ± 1.2 | 1.8 ± 1.1 | **3.07 (2.48–3.80)***** |
| Peer pressure | 1.6 ± 1.0 | 1.7 ± 1.0 | 1.6 ± 0.9 | 1.19 (0.96–1.47) |
| Habit | 1.7 ± 1.0 | 2.2 ± 1.2 | 1.3 ± 0.7 | **2.96 (2.20–3.98)***** |

AOR, adjusted odds ratios; 95% CI, 95% confidence intervals.

*$p < 0.05$

**$p < 0.01$

***$p < 0.001$.

†Based on tertile distributions

‡Scale 1 (totally disagree)– 5 (totally agree).

§Item: I think that I can enjoy sex more when I use drugs.

¶ Item: I think that my fear of acquiring an STI decreases when I use drugs during sex.

## Determinants associated with chemsex: Multivariable analysis

Socio-demographics accounted for 6% of the variance in chemsex with age and SES being independently associated. The inclusion of tobacco use significantly increased the variance accounted for in chemsex ($\Delta R^2$ = 0.06, p<0.001); the model explained 13% of the variance in chemsex; tobacco use remained associated. Entry of the sexual behaviour variables significantly increased the variance accounted for ($\Delta R^2$ = 0.33, p<0.001); the model explained 45% of the variance in chemsex; independently associated were the number of sex partners, group sex and/or fisting, and using online dating platforms for sex. Finally, inclusion of psychosocial determinants significantly increased the variance accounted for in chemsex ($\Delta R^2$ = 0.23, p<0.001); the final model explained 69% of the variance. Variables that remained associated with chemsex in the final multivariable model were attitude and the descriptive norm, tobacco use, multiple sex partners, group sex, and using online dating platforms for sex (Table 3).

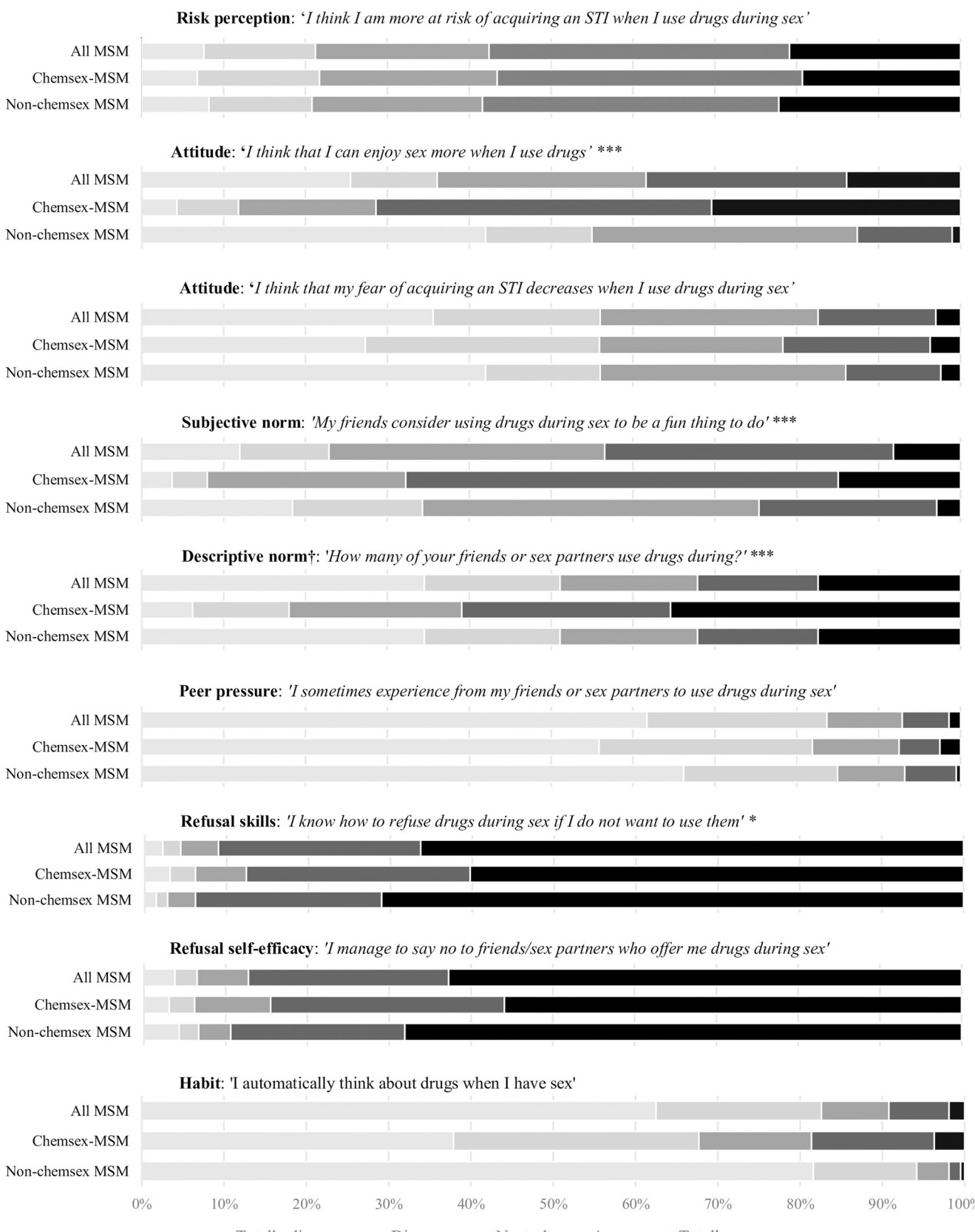

**Fig 2. Proportions of all MSM, MSM who had chemsex, MSM who did not have chemsex reporting specific psychosocial beliefs about chemsex.**
*p < 0.05, **p < 0.01, ***p < 0.001 indicate significant differences between chemsex-MSM and non-chemsex MSM. †Different answering options: No one–everyone.

## Discussion

This is one of the first quantitative studies that explored a broad set of psychosocial, sociodemographic and sexual behavioural determinants for its association with chemsex among MSM. Chemsex was a prevalent behaviour (44% recent practice) in MSM who visited STI clinics in the Netherlands. XTC/MDMA, GHB/GBL and ketamine were the most used drugs during sex. Chemsex was independently associated with believing that the majority of friends and sex partners use drugs during sex (descriptive norm) and believing that sex is more fun when using drugs (attitude). Other determinants were smoking tobacco, having at least five sex partners in the past six months, having group sex and/or fisting, and using online dating platforms. Our conceptual model that included these determinants was able to explain almost seventy percent of the variance in chemsex with psychosocial and sexual behaviour determinants as most important.

In line with previous studies [13,19,20], chemsex was associated with sexual risk behaviour, such as having multiple sex partners and group sex. In addition, we found an independent

**Table 3. Four-step hierarchical regression analysis assessing the impact of categories of determinants on the prediction of chemsex.**

| Step | Variables entered | Step 1 aOR | Step 1 95% CI | Step 2 aOR | Step 2 95% CI | Step 3 aOR | Step 3 95% CI | Step 4 aOR | Step 4 95% CI |
|---|---|---|---|---|---|---|---|---|---|
| 1 | Age | **1.03**** | 1.01–1.05 | **1.03***** | 1.02–1.05 | 1.02 | 0.99–1.04 | 1.01 | 0.98–1.04 |
| | SES, middle | **0.53*** | 0.31–0.90 | **0.57*** | 0.33–0.97 | 0.61 | 0.31–1.18 | 0.78 | 0.34–1.78 |
| | SES, high | 0.83 | 0.50–1.38 | 0.81 | 0.48–1.37 | 0.72 | 0.38–1.39 | 0.90 | 0.40–2.03 |
| 2 | Tobacco smoking, yes | | | **2.62***** | 1.67–4.12 | **2.61**** | 1.51–4.50 | **2.65**** | 1.32–5.32 |
| 3 | Same-sex sexual behaviour, men and women | | | | | 2.04 | 0.82–5.10 | 1.92 | 0.60–6.18 |
| | Number of sex partners six months, 5–10 | | | | | **2.49**** | 1.35–4.61 | **2.69*** | 1.21–6.00 |
| | Number of sex partners six months, > 10 | | | | | 1.65 | 0.76–3.63 | 1.52 | 0.54–4.22 |
| | Number of sex partners during sex, 2 or 3 | | | | | **3.52***** | 1.73–7.15 | 1.58 | 0.64–3.91 |
| | Number of sex partners during sex, > 4 | | | | | 1.07 | 0.40–2.90 | 0.40 | 0.12–1.51 |
| | Group sex, yes | | | | | **5.07**** | 2.02–12.74 | **4.65**** | 1.54–14.05 |
| | Fisting, yes | | | | | **6.53**** | 2.04–20.89 | 2.74 | 0.70–10.65 |
| | Sex at the party scene, yes | | | | | 1.72 | 0.83–3.58 | 1.20 | 0.46–3.15 |
| | HIV status, positive | | | | | **2.95*** | 1.05–8.32 | 2.45 | 0.67–9.00 |
| | Use of online platforms for sex, yes | | | | | **3.10**** | 1.51–6.38 | **2.73*** | 1.13–6.62 |
| 4 | Refusal skills | | | | | | | 0.74 | 0.50–1.09 |
| | Attitude (enjoyment)† | | | | | | | **2.06***** | 1.50–2.84 |
| | Subjective norm | | | | | | | 1.05 | 0.70–1.56 |
| | Descriptive norm | | | | | | | **1.95***** | 1.43–2.65 |
| | Habit | | | | | | | 1.17 | 0.78–1.76 |
| $R^2$ | | 0.064 | | 0.125 | | 0.452 | | 0.685 | |
| $\Delta R^2$ | | 0.064*** | | 0.061*** | | 0.327*** | | 0.233*** | |

AOR, adjusted odds ratios; 95% CI, 95% confidence intervals.

*$p < 0.05$

**$p < 0.01$

***$p < 0.001$.

†Item: I think that I can enjoy sex more when I use drugs.

association between chemsex and smoking tobacco, which was also present in two other studies [14,16]. In the long term, a study has shown that MSM practicing chemsex might be at risk of becoming drug dependent and not being able to return to enjoying sex without chemsex drugs [31]. The risk of dependency might vary per drugs used. For example, crystal meth and GHB/GBL are relatively more likely to cause physical addiction than other drugs [32]. Irrespective of causality, the presence of sexual health risks and possible drug and tobacco use related harms in MSM practicing chemsex confirms the importance of health promotion strategies aimed at minimizing these harms. To identify MSM practicing chemsex in health care and tailor health promotion strategies, it is important to understand their characteristics and beliefs. Our study indicates that MSM practicing chemsex are a heterogeneous group, as chemsex was comparably present in MSM of different age groups, socioeconomic statuses, ethnicities, educational levels and in MSM living in urban and non-urban areas. One study [14] found no differences by educational level, SES, and ethnicity between MSM practicing chemsex and MSM not practicing chemsex. A systematic review [1] found that chemsex participation peaks between the age mid-thirties and early fourties, but is evident at all ages, which was consistent with our study. This suggests that health services, such as STI clinics, should ask about chemsex in all MSM.

In general, MSM practicing chemsex reported to be aware of the STI risks related to drug use during sex and this risk perception did not differ between MSM practicing chemsex and MSM not practicing chemsex. Almost eighty percent reported that they believed to be more at risk of acquiring an STI when using drugs during sex. Nevertheless, MSM practicing chemsex engage more often in sexual risk behaviour and are more often diagnosed with an STI than MSM not practicing chemsex [10,13,14]. This suggests that men practicing chemsex are not naïve to the STI risks of chemsex, but possible lack understanding of how to manage them. Therefore, promoting STI preventive measures, such as pre-exposure prophylaxis (PrEP) to prevent HIV infection and correct condom use, remains highly relevant in this group. A vast majority of MSM practicing chemsex indicated that they had the skills and self-efficacy to refuse drugs during sex when they not want to use them and did not experience peer pressure. Nevertheless, the social environment seems to play an important role in chemsex. Our study shows that MSM practicing chemsex think that the majority of their friends and sex partners also use drugs during sex while MSM not practicing chemsex think that only a minority use drugs during sex. The descriptive norm was independently associated with chemsex. A previous qualitative study also showed that chemsex was perceived as a normalized behaviour among MSM practicing chemsex in South London [19]. It has been suggested [19] that the proportion of MSM practicing chemsex is overestimated by men who practice chemsex. According to the social influence theory, believing that most people in one's social environment perform a certain behaviour can stimulate individuals to practice these behaviours themselves [33]. Another explanation could be provided by the social selection theory. This theory describes that an individual practicing a certain behaviour would change networks to spend more time with others who also practice the same behaviour. A social network approach would be necessary to assess the direction of social influence. Regardless of the direction, it is likely that MSM practicing chemsex operate within a distinct social network where drug use during sex is normalized [34]. Our study shows that the use of online dating platforms for sex is also independently associated with chemsex. Online dating platforms makes it easier to have chemsex at private settings [35]. The use of certain symbols indicating drug use in personal advertisements in these dating platforms [36] probably stimulates selection of sex partners who also use drugs and increases perceptions of ubiquity of chemsex.

A second psychosocial determinant independently associated with chemsex was a positive attitude towards drug use during sex. MSM practicing chemsex more often believed they

could enjoy sex more when using drugs. Drugs are often used to enhance sexual pleasure [6,7], but when sex without drugs is not experienced as pleasurable anymore men face a risk of becoming drug dependent. One study [6] already described that the pleasurable sensations of chemsex form a positive association between drugs and sex that then provides the automatic and reflective motivation to combine sex and drugs in the future. In our study, almost one in five men practicing chemsex reported to automatically think about drugs when having sex. Health care services should pay attention to men whose chemsex behaviour has become an habit to prevent future possible addiction problems. When minimizing the possible harms of chemsex, such as STIs and addiction, health care services should acknowledge the social influence and pleasurable experiences of chemsex to increase acceptability of these strategies.

## Limitations

Our study should be viewed in light of some limitations. Because of the cross-sectional design, we were unable to assess causal relationships between determinants and chemsex. The questionnaire was completed by high-risk MSM, because they were recruited at the STI clinic and participants relatively had more sex partners than the total MSM population (n = 3493) who visited the participating STI clinics during the study period (median 6 vs. 5, p = 0.02). This could have led to an overestimation of the proportion of MSM practicing chemsex but enabled us to assess determinants associated with chemsex. Furthermore, MSM with a younger age (median age 40 vs. 34 years, p<0.001) and a non-western ethnicity (7% vs. 18%, p<0.001) were underrepresented in our study. Ideally, the psychosocial determinants (especially concerning habit formation) would have been measured by multiple validated items. We wanted to increase the response rate by limiting the items in our questionnaire. That is also the reason why more complex items that have been associated with chemsex in a previous study [29], such as internalized homophobia and minority stress, were not measured in our questionnaire. Furthermore, recall bias and under-reporting bias on some psychosocial beliefs, such as experienced peer pressure to use drugs during sex, could have been possible. Finally, multiple comparisons might have increased risk for type 1 errors. Yet, for exploratory studies, correcting multiple comparisons is often a too strict approach (leading to more type II errors) and a flexible approach, as we took in our study, is deemed more suitable [37].

## Conclusion

Our study shows that MSM practicing chemsex are a heterogeneous group concerning socio-demographic characteristics. STI risk factors, such as having multiple sex partners and group sex, and smoking tobacco were independently associated with chemsex, indicating a need for health promotion in MSM practicing chemsex. MSM practicing chemsex generally believed that the majority of their friends or sex partners also used drugs during sex and this was independently associated with chemsex. This suggests that MSM practicing chemsex are likely to find themselves in distinct social and sexual networks where it is the norm to use drugs when having sex. MSM practicing chemsex generally believed that sex with drugs is more fun and this was also independently associated with chemsex. Health services should acknowledge the social influence and pleasurable experiences to increase acceptability of strategies aimed at minimizing the possible harms of chemsex, such as STIs and addiction.

## Supporting information

**S1 File. Informed consent in Dutch and English.**
(DOCX)

**S2 File. List of participating STI clinics.**
(DOCX)

**S3 File. Online questionnaire in Dutch and English.**
(DOCX)

## Acknowledgments

The authors wish to acknowledge the support of Leon Knoops of Mainline and Arthur Kleisterlee of Mondriaan for their help in developing the questionnaire. We greatly thank all the staff of the participating STI clinics for the recruitment of our participants, and in particular Marga Smit, Mandy Sanders, Mariska Muyrers, Luuk Levels, Karlijn Kampman, Sophie Kuizenga-Wessel, Marie-Sophie Mutsaers, Nienke Bakker, Helmie van der Meijden, Decontee Shilue and Harriette van Buel. We also would like to thank all participants for completing the questionnaire.

## Author Contributions

**Conceptualization:** Ymke J. Evers, Jill J. H. Geraets, Geneviève A. F. S. Van Liere, Christian J. P. A. Hoebe, Nicole H. T. M. Dukers-Muijrers.

**Data curation:** Ymke J. Evers, Jill J. H. Geraets.

**Formal analysis:** Ymke J. Evers, Jill J. H. Geraets.

**Methodology:** Ymke J. Evers, Jill J. H. Geraets, Geneviève A. F. S. Van Liere, Christian J. P. A. Hoebe, Nicole H. T. M. Dukers-Muijrers.

**Project administration:** Ymke J. Evers.

**Supervision:** Ymke J. Evers, Geneviève A. F. S. Van Liere, Christian J. P. A. Hoebe, Nicole H. T. M. Dukers-Muijrers.

**Visualization:** Ymke J. Evers, Jill J. H. Geraets.

**Writing – original draft:** Ymke J. Evers, Jill J. H. Geraets, Geneviève A. F. S. Van Liere, Christian J. P. A. Hoebe, Nicole H. T. M. Dukers-Muijrers.

**Writing – review & editing:** Ymke J. Evers, Jill J. H. Geraets, Geneviève A. F. S. Van Liere, Christian J. P. A. Hoebe, Nicole H. T. M. Dukers-Muijrers.

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
