## [Decision Letter · Decision Letter 0]

21 Apr 2020

PONE-D-19-31468

Attitude and beliefs about the social environment associated with chemsex among MSM visiting STI clinics in the Netherlands: an observational study

PLOS ONE

Dear Mrs Evers,

Thank you for submitting your manuscript to PLOS ONE. After careful consideration, we feel that it has merit but does not fully meet PLOS ONE’s publication criteria as it currently stands. Therefore, we invite you to submit a revised version of the manuscript that addresses the points raised during the review process.

From my own reading of the manuscript, I agree with the reviewer’s comments. Please carefully consider their suggestions. I look forward to receiving your revision. 

We would appreciate receiving your revised manuscript by Jun 05 2020 11:59PM. To enhance the reproducibility of your results, we recommend that if applicable you deposit your laboratory protocols in protocols.io, where a protocol can be assigned its own identifier (DOI) such that it can be cited independently in the future. For instructions see: http://journals.plos.org/plosone/s/submission-guidelines#loc-laboratory-protocols

We look forward to receiving your revised manuscript.

Kind regards,

Ethan Morgan

Academic Editor

PLOS ONE

Journal Requirements:

2. Please refer to any sample size calculations or post-hoc corrections to correct for multiple comparisons during your statistical analyses. If these were not performed please justify the reasons. Please refer to our statistical reporting guidelines for assistance (https://journals.plos.org/plosone/s/submission-guidelines.#loc-statistical-reporting).

3.  Please provide additional details regarding participant consent. In the ethics statement in the Methods and online submission information, please ensure that you have specified how verbal consent was documented and witnessed.

4.  Please include additional information regarding the survey or questionnaire used in the study and ensure that you have provided sufficient details that others could replicate the analyses. For instance, if you developed a questionnaire as part of this study and it is not under a copyright more restrictive than CC-BY, please include a copy, in both the original language and English, as Supporting Information. In addition, please include details of any pretesting of this questionnaire, including the number and nature of the participants.

Thank you for your attention to these queries.

Reviewers' comments:

Reviewer's Responses to Questions

**Comments to the Author**

1. Is the manuscript technically sound, and do the data support the conclusions?

Reviewer #1: Partly

Reviewer #2: Yes

2. Has the statistical analysis been performed appropriately and rigorously? 

Reviewer #1: No

Reviewer #2: Yes

3. Have the authors made all data underlying the findings in their manuscript fully available?

Reviewer #1: Yes

Reviewer #2: Yes

4. Is the manuscript presented in an intelligible fashion and written in standard English?

Reviewer #1: Yes

Reviewer #2: Yes

5. Review Comments to the Author

Reviewer #1: This paper addresses an important subject: the association between drug use for sexual pleasure and HIV infection among men who have sex with men. The paper is reasonably well written but there are a number of conceptual and analytical problems that would need to be addressed before consideration for publication. They are listed below.

1) Although the authors are correct that chemsex may involve a variety of substances, historically crystal meth has been identified and studied as the core and most problematic substance involved in chemsex. In the present study, 85% of MSM reports to use XTC or MDMA for chemsex, of which the short and long term effects are very different from crystal meth (reported only by 9%). The authors should describe these differences in terms of fysiology, psychosocial and psychopathological effects and risks for dependency. They should also keep it in mind when they discuss and interprete their findings. Since most other studies aim to explain and understand crystal meth use in relation to HIV infection and other sequelae, this may affect the comparability of the present study within the total body of literature on the subject.

2) In their analysis, the authors merge group sex and "fisting" in one variable labeled "esoteric sexual practices". This variable should be disaggregated and factors reported and analyzed seperately. Group sex is not esoteric and one of the main features of chemsex and unlike "fisting" a strong independent predictor of HIV acquisition. It should not be lumped together with something of a complete different order. This reviewer also argues that group sex should not be seen/analyzed as a determinant of chemsex, but rather as an effect or something occurring downstream in the causal chain.

3) The authors define their psychosocial variable "habit" as "thinking about drugs when having sex". This is not how the habit forms itself. It is the other way around. When people think, fantasize or plan for sex, they think about drugs. Sex implies drugs and craving happens when thinking about it. Hence the thinking (and preparation) happens before and not during sex. It may be better to remove this variable from the model or clearly address its limitations in the methods or discussion section.

4) Non-response was considerable in the present study (53%). The authors should use data available from the electronic patient registry to compare responders with non-responders to assess potential bias and comment on the implications when generalizing and interpreting the results.

Reviewer #2: The study provides important findings about MSM practicing chemsex. A few comments for your consideration:

Major Comments:

• The conclusion of “heterogeneous group regarding socio-demographics” is unlikely to be supported by the insignificant difference of socio-demographics between the two groups. Insignificant difference, from my point of view, simply shows that the background of the two groups are similar.

• In DISCUSSION, paragraph 2, “the clustering of sexual health risks and possible drug and tobacco use” is unlikely to be supported by the findings. No cluster analysis was performed in this study. Please consider if you want to modify the description.

Minor Comments:

• ABSTRACT line 30, please indicate clearly that the study population referred to those having sex in the past 6 months.

• INTRODUCTION lines 97-98: please move Figure 1 (and its citation) to METHODS

• METHODS: Different in-text citation methods are observed, please standardise, e.g. line 111 compared with lines 134-135

• References in lines 178-179 are missing

• RESULTS: I am wondering if there are any multidrug data

• Mismatch between RESULTS and TABLE 2. In main text, the psychological determinants are described in terms of median, while TABALE 2 shows mean and SD. Interestingly, the number is the same. Please clarify whether they are median and IQR or mean and SD.

6. PLOS authors have the option to publish the peer review history of their article (what does this mean?). If published, this will include your full peer review and any attached files.

Reviewer #1: No

Reviewer #2: No

---

## [Author Response · Author response to Decision Letter 0]

14 May 2020

Dear editor and reviewers,

Thank you for giving us the opportunity to revise our manuscript entitled: ‘Attitude and beliefs about the social environment associated with chemsex among MSM visiting STI clinics in the Netherlands: an observational study’. 

We greatly thank the editor and reviewers for their helpful and constructive comments, all addressed in our point-by-point reply (in the uploaded file). We revised and improved the manuscript accordingly. We are very pleased with the improvements made and hope you will find them satisfactory.

Kind regards, also on behalf of the co-authors,

Ymke Evers

---

## [Decision Letter · Decision Letter 1]

3 Jun 2020

PONE-D-19-31468R1

Attitude and beliefs about the social environment associated with chemsex among MSM visiting STI clinics in the Netherlands: an observational study

PLOS ONE

Dear Dr. Evers,

Thank you for submitting your manuscript to PLOS ONE. After careful consideration, we feel that it has merit but does not fully meet PLOS ONE’s publication criteria as it currently stands. Therefore, we invite you to submit a revised version of the manuscript that addresses the points raised during the review process.

As per the reviewer's suggestions, please take care to ensure all remnants of previous methods are removed.

We look forward to receiving your revised manuscript.

Kind regards,

Ethan Morgan

Academic Editor

PLOS ONE

Reviewers' comments:

Reviewer's Responses to Questions

**Comments to the Author**

1. If the authors have adequately addressed your comments raised in a previous round of review and you feel that this manuscript is now acceptable for publication, you may indicate that here to bypass the “Comments to the Author” section, enter your conflict of interest statement in the “Confidential to Editor” section, and submit your "Accept" recommendation.

Reviewer #1: (No Response)

Reviewer #2: All comments have been addressed

2. Is the manuscript technically sound, and do the data support the conclusions?

Reviewer #1: Yes

Reviewer #2: Yes

3. Has the statistical analysis been performed appropriately and rigorously? 

Reviewer #1: Yes

Reviewer #2: I Don't Know

4. Have the authors made all data underlying the findings in their manuscript fully available?

Reviewer #1: Yes

Reviewer #2: Yes

5. Is the manuscript presented in an intelligible fashion and written in standard English?

Reviewer #1: Yes

Reviewer #2: Yes

6. Review Comments to the Author

Reviewer #1: The authors have done a great job in addressing the reviewer comments given the limitation of their data.

Even though they have disaggregated group sex and fisting in their analysis (leading to much more plausible results), some sediments of the initial approach can be found in line 163-164 (of the revised version) in the methods section where they are still talking about lumping these together as one"esoteric practice". This needs to be corrected.

In line 295 (revised version) the authors introduce a new concept called "sober sex" as opposed to chemsex. Since not all drugs are included in the definition of chemsex, sober sex is not necessarily sober, but rather without sexual stimulant drugs. The statement should be changed accordingly.

Reviewer #2: (No Response)

7. PLOS authors have the option to publish the peer review history of their article (what does this mean?). If published, this will include your full peer review and any attached files.

Reviewer #1: Yes: Frits van Griensven

Reviewer #2: No

---

## [Author Response · Author response to Decision Letter 1]

15 Jun 2020

Dear editor and reviewers,

Thank you for giving us the opportunity to revise our manuscript entitled: ‘Attitude and beliefs about the social environment associated with chemsex among MSM visiting STI clinics in the Netherlands: an observational study’. 

We greatly thank the editor and reviewer for their helpful comments, all addressed below in our point-by-point reply. We are pleased with the improvements and hope you will find them satisfactory. 

Kind regards, also on behalf of the co-authors,

Ymke Evers

 

Reviewer #1:

The authors have done a great job in addressing the reviewer comments given the limitation of their data.

Response: Thank you for the constructive feedback.

1. Even though they have disaggregated group sex and fisting in their analysis (leading to much more plausible results), some sediments of the initial approach can be found in line 163-164 (of the revised version) in the methods section where they are still talking about lumping these together as one"esoteric practice". This needs to be corrected.

Response: We corrected this as suggested by the reviewer and deleted the word esoteric practice. 

2. In line 295 (revised version) the authors introduce a new concept called "sober sex" as opposed to chemsex. Since not all drugs are included in the definition of chemsex, sober sex is not necessarily sober, but rather without sexual stimulant drugs. The statement should be changed accordingly.

Response: We changed this sentence as suggested by the reviewer: ‘…that MSM practicing chemsex might be at risk of becoming drug dependent and not being able to return to enjoying sex without chemsex drugs’.

---

## [Editor Report · Decision Letter 2]

17 Jun 2020

Attitude and beliefs about the social environment associated with chemsex among MSM visiting STI clinics in the Netherlands: an observational study

PONE-D-19-31468R2

Dear Dr. Evers,

We’re pleased to inform you that your manuscript has been judged scientifically suitable for publication and will be formally accepted for publication once it meets all outstanding technical requirements.

Kind regards,

Ethan Morgan

Academic Editor

PLOS ONE
---

## [Editor Report · Acceptance letter]

22 Jun 2020

PONE-D-19-31468R2 

Attitude and beliefs about the social environment associated with chemsex among MSM visiting STI clinics in the Netherlands: an observational study 

Dear Dr. Evers:

I'm pleased to inform you that your manuscript has been deemed suitable for publication in PLOS ONE. Congratulations! Your manuscript is now with our production department. 

Kind regards, 

on behalf of

Dr. Ethan Morgan 

Academic Editor

PLOS ONE